# Recent Progress and Challenges in the Development of Antisense Therapies for Myotonic Dystrophy Type 1

**DOI:** 10.3390/ijms232113359

**Published:** 2022-11-01

**Authors:** Thiéry De Serres-Bérard, Siham Ait Benichou, Dominic Jauvin, Mohamed Boutjdir, Jack Puymirat, Mohamed Chahine

**Affiliations:** 1CERVO Research Center, Institut Universitaire en Santé Mentale de Québec, Quebec City, QC G1J 2G3, Canada; 2LOEX, CHU de Québec-Université Laval Research Center, Quebec City, QC G1J 1Z4, Canada; 3Cardiovascular Research Program, VA New York Harbor Healthcare System, New York, NY 11209, USA; 4Department of Medicine, Cell Biology and Pharmacology, State University of New York Downstate Health Science University, New York, NY 11203, USA; 5Department of Medicine, NYU School of Medicine, New York, NY 10016, USA; 6Department of Medicine, Faculty of Medicine, Université Laval, Quebec City, QC G1V 0A6, Canada

**Keywords:** antisense oligonucleotides, myotonic dystrophy type 1, myotonic dystrophy protein kinase, alternative splicing

## Abstract

Myotonic dystrophy type 1 (DM1) is a dominant genetic disease in which the expansion of long CTG trinucleotides in the 3′ UTR of the myotonic dystrophy protein kinase (*DMPK*) gene results in toxic RNA gain-of-function and gene mis-splicing affecting mainly the muscles, the heart, and the brain. The CUG-expanded transcripts are a suitable target for the development of antisense oligonucleotide (ASO) therapies. Various chemical modifications of the sugar-phosphate backbone have been reported to significantly enhance the affinity of ASOs for RNA and their resistance to nucleases, making it possible to reverse DM1-like symptoms following systemic administration in different transgenic mouse models. However, specific tissue delivery remains to be improved to achieve significant clinical outcomes in humans. Several strategies, including ASO conjugation to cell-penetrating peptides, fatty acids, or monoclonal antibodies, have recently been shown to improve potency in muscle and cardiac tissues in mice. Moreover, intrathecal administration of ASOs may be an advantageous complementary administration route to bypass the blood-brain barrier and correct defects of the central nervous system in DM1. This review describes the evolution of the chemical design of antisense oligonucleotides targeting CUG-expanded mRNAs and how recent advances in the field may be game-changing by forwarding laboratory findings into clinical research and treatments for DM1 and other microsatellite diseases.

## 1. Introduction

RNA-based therapies have emerged in the last decades as a promising avenue for regulating various molecular pathways, opening the door for treating diseases that otherwise would be difficult to target with conventional small molecules and recombinant proteins. The development of therapies based on messenger RNAs (mRNAs), RNA aptamers, small interfering RNAs (siRNAs), and antisense oligonucleotides (ASOs) to treat multiple disorders have been the subject of extensive investigations. After several years of slower development, new strategies to enhance the cellular uptake of RNA therapeutics and their resistance to nuclease degradation have boosted their potency and relevance for clinical use. ASOs are short single-stranded sequences of nucleic acid, typically between 15–25 bases long, which are chemically modified to improve their stability and specificity. By hybridizing with their complementary RNA targets, ASOs modulate their processing or their expression, usually by inhibiting their interactions with spliceosomes, ribosomes, and RNA-binding proteins (RBPs) or by promoting their enzymatic degradation. To date, 13 ASO and siRNA drugs have been approved by the U.S. Food and Drug Administration (FDA) for clinical use [1,2], including nusinersen (Spinraza) for treating spinal muscular atrophy (SMA) [3,4], as well as eteplirsen (Exondys 51), golodirsen (Vyondys 53), casimersen (Amondys 45) and viltolarsen (Viltepso) for treating Duchenne muscular dystrophy (DMD) [5,6,7,8].

Myotonic dystrophy type 1 (DM1) is a common muscular dystrophy involving a toxic RNA gain-of-function resulting from the expansion of CTG trinucleotides in the 3′ UTR of the myotonic dystrophy protein kinase (*DMPK*) gene [9]. The CUG-expanded transcripts adopt stable RNA hairpin conformations and are retained in the nucleus in distinct foci [10]. Splicing factors from the muscleblind-like protein (MBNL) family have a high affinity for CUG repeats and are sequestered by the mutant *DMPK* mRNAs, impairing their functions, and causing the expression of embryonic splice isoforms in adult tissues [11]. This effect is further aggravated by altered phosphorylation and increased expression of RBPs from the CUGBP Elav-like family (CELF) [12,13,14]. Various mis-splicing events have been associated with a wide range of symptoms manifested by DM1 patients, notably impaired relaxation of muscles (myotonia), muscle weakness, cardiac conduction defects, and cognitive deficits. These symptoms have been recapitulated in different mice models expressing transcripts with expanded CUG repeats, confirming that the toxic RNA gain-of-function is a central mechanism in the disease [15,16,17]. DM1 thus constitutes an ideal candidate for developing pioneer ASO platforms for diseases involving mRNA dysregulation and other microsatellite disorders. In 2014, Ionis Pharmaceuticals initiated the first phase 1/2a DM1 clinical trial based on an antisense strategy (NCT02312011). Ionis reported that subcutaneous administration of multiple escalating doses of the chemically modified IONIS-DMPKRx ASO in adults with DM1 was well tolerated. However, despite inducing minor positive changes in biomarkers and splicing at higher dosages, the ASO concentration measured in muscle biopsies did not reach the therapeutic value of ~10 µg/mg required to support further clinical investigations [18]. Ionis Pharmaceuticals instead announced that it would focus on the development of ligand-conjugated ASOs (LICAs) that bind to specific cell receptors and increase drug concentration and potency in targeted tissues. In recent years, the linkage of ASOs with molecules that facilitate cellular uptake has indeed been an important area of research.

Here, we describe the chemical designs of ASOs and the conjugates investigated in preclinical DM1 mouse models. A particular focus is placed on improving drug uptake in the muscles, the heart, and the brain, which are the main organs affected by the disease. We also discuss the factors impacting the safety of antisense therapy and the potential of ASOs for shaping future treatments for myotonic dystrophy and other microsatellite diseases.

## 2. Evolution of the Chemistries and Pharmaceutical Properties of ASOs

### 2.1. First-Generation ASOs

Antisense therapy is an attractive strategy making it possible to target almost every gene by simply modifying the base sequence. However, unmodified nucleic acid oligos are highly sensitive to nucleases and have a limited half-life in vivo. Therefore, various chemical designs have been investigated to improve ASOs’ activity and tolerability (Figure 1). In the first generation of chemically modified ASOs, the phosphate backbone of nucleic acids is substituted by phosphorothioate (PS) bonds where a non-bridging oxygen is replaced by a sulfur atom. It has been known for about 50 years now that this modification makes the oligonucleotides resistant to nuclease degradation, likely because of the larger van der Waals radius of the sulfur atom displacing the metal ions at the active sites of nucleases [19,20,21]. Furthermore, PS-modified ASOs enable the cleavage of the hybridized RNA strand by RNase H [22,23]. As the ASO is still active following target degradation, this mechanism dramatically enhances its inhibitory activity. ASOs synthesized with PS modifications also display an increased affinity for plasma proteins, notably albumin, likely because of interactions between the anionic sulfur and cationic amino acids. ASO binding to serum proteins facilitates cell uptake via the endosomal pathway and slows their excretion in urine following glomerular filtration [24]. However, PS-ASO interactions with paraspeckle proteins may be hepatotoxic [25]. Alternatively, the phosphate backbone may be replaced by N3′-->P5′ phosphoramidate linkages in which the 3′ oxygen of the furanose ring is replaced by a 3′-amino group. Phosphoramidate ASOs are resistant to nucleases and have a high affinity for single-strand RNAs [26,27]. However, they do not activate the RNase H pathway and thus act by steric hindrance to inhibit translation initiation and interactions with RBPs without directly altering the target RNA.

### 2.2. Second-Generation ASOs

To overcome the limitations of PS modifications, second-generation ASOs were generated by adding alkyl groups to the 2′ hydroxyl of the sugar moiety. Notably, 2′-O-methyl (2′-O-Me) and 2′-O-methoxyethyl (2′-MOE) modifications improve both RNA hybridization and resistance to nucleases while also being less toxic than ASOs modified with PS bonds [28,29,30]. Oligonucleotides with only 2′-alkyl modifications are not recognized by RNase H. Given this, chimeric ASOs called gapmers are often synthesized with a PS or unmodified central gap of deoxyribonucleotide compatible with RNase H activity and are combined with terminal 2′-alkyl modifications to improve their stability. Increasing the RNA hybridization properties of ASOs is especially important to efficiently target RNAs that form secondary structures and that are less accessible to unmodified ASOs, as is the case with CUG-expanded mRNAs [31]. 

### 2.3. Third-Generation ASOs

To further improve their pharmacokinetic properties, third-generation ASOs have been designed by modifying the furanose ring. Locked nucleic acids (LNAs) are generated by tethering the 2′ residue of the sugar moiety with the carbon atom at position 4′. The resulting reduced conformational flexibility of the ribose enhances affinity binding to RNAs and may increase the potency 5 to 10-fold with respect to 2′-MOE gapmers [32,33]. Alternatively, the addition of an extra methylene unit in the bridge connecting the 2′ oxygen and the 4′ carbon led to the generation of 2′-O,4′-C-ethylene-bridged nucleic acid (ENA) with better nuclease resistance and RNA affinity properties than the LNA chemistry [34]. LNA ASOs have been reported to be hepatotoxic, limiting their potential for clinical use [33,35,36,37]. A compromise between the LNA and 2′-MOE chemistries has led to the generation of a 2′,4′-constrained ethyl (cEt) modification in which the 2′-O-ethyl substitute is linked to the 4′ position. ASOs with cEt modifications retain a high affinity for RNAs without causing overt toxicity [38,39]. Another LNA analog has been designed by adding an N-O bond to the bridge of the sugar moiety to form a 2′,4′-aminomethylene bridged nucleic acid (2′,4′-BNA^NC^) [40]. The additional nitrogen atom increases ASO target affinity by lowering the repulsion with the negatively charged backbone. The nitrogen atom can also act as a conjugating site to further modify the properties of the ASO by incorporating a hydrogen atom (2′,4′-BNA^NC^[NH]) or a methyl group (2′,4′-BNA^NC^[NME]), which may improve nuclease resistance and target discrimination [40,41]. Substitution of the furanose ring with a morpholine ring combined with phosphorodiamidate modification of the backbone of the oligomer (PMO) is another chemical modification that has been used in a clinical setting. PMO ASOs are neutrally charged, reducing the possibility of interactions with proteins while displaying high resistance to nucleases and enhanced stability [42]. Since PMOs do not activate the RNase H pathway, they are used notably as splice-switching ASOs for treating diseases such as SMA and DMD [43,44]. Another category of nucleotide analog, peptide nucleic acid (PNA), is generated by substituting the phosphate backbone with a pseudopeptide backbone of N-(2-aminoethyl)-glycine linked to the nucleobases by methylene carbonyl [45,46]. PNAs are resistant to nucleases but retain a geometry like that of oligonucleotides and can form stable duplexes with DNA and RNA by Watson–Crick pairing [47]. They have a neutral charge and, thus, bind to RNAs and DNAs with high affinity due to the lack of electrostatic repulsions. As PNAs are randomly folded, combining them with a hydrophilic (R)-diethylene glycol unit (miniPEG) on the γ-backbone of PNAs (MPγPNAs) gives them a right-handed helix conformation, improving both their solubility in water and their affinity for nucleic acids [48]. PNAs do not activate RNase H, and thus, mainly act on their targets by steric hindrance.

### 2.4. Pharmacokinetics and Cell Distribution of ASOs

Chemical modifications of ASOs are an important factor that determines their pharmacokinetic properties. ASOs are usually systematically administrated by subcutaneous or intravenous injection, absorbed in the plasma, and distributed quickly to tissues, except for the central nervous system (CNS) [49]. Higher ASO concentrations usually accumulate in the liver and the kidneys due to their fenestrated and sinusoidal capillaries, in contrast with cardiac and skeletal muscle tissues, which have a continuous endothelium. The pharmacokinetics of ASOs are characterized by a rapid initial clearance phase from the plasma that occurs over a few hours followed by a slow terminal elimination phase of ASOs absorbed in tissues, which takes place over a few days to several weeks. Negatively charged ASOs, which bind to serum proteins, are more slowly excreted by the kidneys than uncharged ASOs such as PNAs and PMOs. Therefore, these latter usually require conjugates or being administrated at higher doses to be absorbed by tissues at sufficient levels. ASOs are thought to penetrate in myocytes mainly following receptor binding and clathrin-mediated endocytosis, although ASOs’ internalization by clathrin-independent processes may also occur [49,50,51,52]. Only a small portion of ASOs take part in a productive pathway by escaping from endocytic vesicles and shuttling to the nucleus through the nuclear pores, where expanded *DMPK* mRNAs are localized [52,53]. Most ASOs are retained in endosomes and lysosomes, where they are ultimately degraded and considered non-productive [52]. ASO concentrations in the nucleus in the nanomolar range are required to achieve significant therapeutic effects, with a lower concentration for RNA therapeutics with catalytic activity than for those acting by steric hindrance [54]. Predominant distribution of ASOs with ENA modifications in vesicle-like structures instead of the nucleus in myoblasts has been associated with poor silencing of CUG-expanded *DMPK* transcripts, in contrast with other chemistries [55]. 

## 3. In Vitro Models of DM1 for ASOs Screening

In addition to the chemical design, ASO’s potency also depends on the presence of secondary structures and functional motifs in the target mRNA as well as the thermodynamic stability of ASO and RNA duplexes [56]. Optimal target sequences can be determined using computational algorithms to increase the hit rate of newly designed ASOs with an adequate potency and safety profile [57,58,59,60,61,62]. Alternatively, the mRNA walking method consisting of testing a high number of ASOs covering the entire target transcript is almost guaranteed to identify the optimal sequences but is highly labor extensive and used mainly by the industry. Notably, IONIS screened hundreds of cEt-modified gapmer ASOs in human skeletal muscle cells to identify sequences leading to *DMPK* mRNA knockdown, followed by in vivo evaluation of their tolerability in mice and rats [63]. These results led to the identification of IONIS 486178 (or ISIS 486178), which has been extensively investigated in DM1 mice models [64,65,66,67]. In vitro screening is an important step in the development of antisense therapy, which requires accurate models to identify the best drug candidates. 

Muscle stem cells (or satellite cells) are found in a quiescent state between the sarcolemma and the basal lamina of mature skeletal muscles. Upon activation, following injury or intense exercises, satellite cells differentiate into proliferative myoblasts and fuse to form multinucleated myotubes. Satellite cells have been successfully derived from DM1 muscle biopsies and differentiated into myotubes in vitro, although showing delayed maturation and fusion defects when derived from DM1 fetal tissues [68,69,70]. However, the limited proliferative capacity and progressive senescence with time in culture limit the potential of this model for massive ASO screening. To overcome this limitation, primary DM1 myoblasts have been immortalized by inducing ectopic expression of the human telomerase reverse transcriptase (TERT) in combination with the cell cycle drivers cyclin D1 (CCND1) and cyclin-dependent kinase 4 (CDK4) [71,72]. In contrast with other immortalization processes relying on viral oncogenes, this method allows myoblasts to completely exit the cell cycle and undergo efficient myogenic differentiation. Reprogramming of human somatic cells, such as skin fibroblasts or blood and urine-derived cells, into induced pluripotent stem cells (hiPSCs) also provides a source of DM1 cells with high proliferative potential, which can be obtained by less invasive methods than muscle biopsy. Alternatively, DM1 myoblasts have also been transdifferentiated through direct reprogramming of skin fibroblasts following lentiviral-induced MyoD expression [73]. Generation of DM1 cell lines from patients with different clinical forms of DM1, such as the congenital, infantile, adult-onset, or late-onset forms can be advantageous for assessing drug efficacy on different phenotypes of the disease. Treatments of DM1 myotubes derived from immortalized myoblasts, MyoD-induced fibroblasts, or hiPSCs with (CAG)7 ASOs with PS and 2′-O-Me modifications have been shown to successfully decrease foci formation and, in some cases, partially reverse mis-splicing defects, supporting the utility of these models for ASOs screening [71,72,73,74]. Differentiation of infantile DM1 hiPSCs into neural progenitor cells has also been achieved to validate the use of IONIS 486178 for rescuing MBNL sequestration by foci and neuronal alternative splicing defects [65]. Since splicing dysregulation in DM1 mainly involve biased production of fetal isoforms in adult tissues, insufficient maturation of in vitro differentiated cells may hinder the detection of alternative splicing correction by ASOs [72,75]. Therefore, optimized protocols to yield mature cells following stem cell differentiation are still needed to use alternative splicing as a reliable biological readout for ASO screening. Nonetheless, in vitro models are a powerful tool to identify promising ASO candidates to be evaluated in preclinical mice models of DM1. 

## 4. Mice Models of RNA Toxicity for the Investigation of Antisense Therapy

### 4.1. The HSA^LR^ Mouse Model

The central role of the toxic RNA gain-of-function in myotonic dystrophy has been validated using different transgenic mouse models expressing transcripts with CUG expansions. Notably, a transgenic mouse model carrying the human skeletal actin gene (*ACTA1)* with long CTG repeats (HSA^LR^ mouse) is commonly used for preclinical studies because it expresses high levels of *ACTA1* transcripts with 250 CUG repeats in the 3′ UTR, causing the sequestration of MBNL in myocytes as well as severe myotonia and progressive myopathy [15]. CUG-expanded skeletal α-actin mRNAs are expressed over 1000 times more than the endogenous mouse *Dmpk* or *DMPK* mRNAs in human muscles [76], setting the bar high for identifying ASOs with enough potency to achieve significant results in clinical trials. The numerous mis-splicing alterations caused by CUG-expanded RNAs offer a wide array of biomarkers for drug screening [77]. For assessing ASO potency with a non-invasive imaging method, a therapy reporter (TR; HSA^LR^) bi-transgenic mouse model expressing both *ACTA1* transcripts with more than 200 CUG repeats and a bi-chromatic human *ATP2A1* exon 22 reporter gene under the control of an HSA promoter has been recently generated [78]. The inclusion of the *ATP2A1* minigene exon 22 in wild-type mice produces a red fluorescent protein (DsRed) while the exclusion of exon 22, which is a robust mis-splicing event sensible to ASO therapy in adult HSA^LR^ mice, causes a reading frameshift leading to the production of a green fluorescent protein (GFP). However, the HSA^LR^ mouse is restricted to investigating ASOs that target *ACTA1* transcripts or the CUG trinucleotides in muscles, as the transgene does not contain the human *DMPK* sequence.

### 4.2. The DMSXL Mouse Model

In an effort to induce a *DMPK* expression pattern more representative of patients, another model, the DM300 mouse, has been modified with a transgene containing a 45-kb length of a human DM1 locus with >300 CTG repeats [79]. Given the instability of the microsatellite tract, which is biased toward expansion, successive breeding was performed to generate DMSXL mice that express human *DMPK* mRNAs with more than 1000 CUG repeats [16]. Intriguingly, homozygous DMSXL mice develop symptoms reminiscent of patients with a high number of CTG repeats. Indeed, mutations exceeding 1000 CTG repeats in patients often lead to the development of congenital or infantile-onset forms of DM1 associated with hypotonia and intellectual disabilities with the absence of myotonia during childhood [80]. Similarly, DMSXL mice exhibit only mild myotonia at older ages but have reduced neonatal survivability, developmental defects, muscle weakness, abnormal behavior, and synaptic dysregulation [81]. DMSXL mice also manifest electrical and contractile cardiac dysfunctions following exposure to a sodium channel blocker, with a more pronounced phenotype in older mice [82]. DMSXL mice feature mild mis-splicing alterations in muscles compared to HSA^LR^ mice, which may be explained by the low expression of the CUG-expanded *DMPK* transcripts (10-fold lower than endogenous mice *Dmpk* mRNAs) [76]. The DMSXL mouse is thus useful for modeling the efficacy of ASOs for treating early-onset forms of DM1 associated with large CUG expansions as well as for investigating their effects on brain alterations. 

### 4.3. Inducible Mouse Models of RNA Toxicity

To create conditional RNA toxicity, another model, the DM200 mouse, was generated by the addition of a GFP-*DMPK* 3′ UTR transgene with more than 200 CTG repeats under the control of a tetracycline-inducible *DMPK* promoter, and a transgene encoding the reverse tetracycline transactivator under the control of the ubiquitous CMV promoter [17]. This system allows the reversible expression of the CUG-expanded transcripts following exposure to doxycycline in both the heart and muscles. Induced DM200 mice feature myotonia and important cardiac conduction abnormalities, such as progressive heart block and atrioventricular node dysfunction. Stopping doxycycline administration in these mice reverses myotonia and mis-splicing in addition to stabilizing or improving heart conduction defects, indicating that reducing CUG-expended RNAs with antisense therapy is a viable strategy for treating myotonic dystrophy. To allow the inducible expression of RNAs with large CUG expansions specifically in the skeletal muscles, a Cre-lox system was generated by crossing mice carrying the human *DMPK* exon 15 containing 960 CTG repeats (EpA960) and a floxed polyadenylation cassette with mice having a tamoxifen-inducible Cre-ERT2 recombinase transgene under the control of an HSA promoter (HSA-Cre-ERT2) [83]. Despite expressing fewer CUG-expanded RNAs than HSA^LR^ mice, EpA960/HSA-Cre-ERT2 mice exposed to tamoxifen manifest CELF1 overexpression and more severe muscle wasting than the latter in addition to myotonia. Similarly, the EpA960/MCM mouse model, in which the tamoxifen-inducible MerCreMer (MCM) is under the control of the α–myosin heavy chain (α-MHC) promoter, allows the inducible expression of RNAs with 960 CUG repeats specifically in the heart. This model recapitulates arrhythmias as well as systolic and diastolic dysfunctions, making it useful for investigating cardiac defects [84]. EpA960/CaMKII-Cre mice were also generated to allow post-natal induction of RNA toxicity in the brain, allowing it to recapitulate cortical atrophy, corpus callosum thinning and neurites degenerations [85]. Globally, the symptoms and molecular features developed by the different DM1 mouse models seem to depend on the tissue-specific expression level of the transgene during development as well as the number of CTG repeats. This is coherent with the surprising heterogeneity in the clinical presentation of patients with DM1. Preclinical studies on different mouse models may be required to evaluate the effects of ASOs on various parameters involved in the disease, such as aberrant splicing, myotonia, muscle weakness and wasting, neurological defects, and cardiac pathology. 

## 5. Targeting *DMPK* mRNAs with Antisense Therapy

### 5.1. Steric Blocking ASOs to Restore MBNL Function

Due to the major contribution of RNA toxicity to DM1 pathology, most antisense strategies aim to target CUG-expanded *DMPK* transcripts (Table 1). Our group showed more than two decades ago that viral transduction to express antisense RNAs complementary to the CUG repeats in human DM1 myoblasts decreases the level of mutant *DMPK* transcripts and corrects CELF1 protein levels [86]. The two main strategies used for treating DM1 with antisense therapy consist of causing the degradation of CUG-expanded RNAs, usually via the RNase H pathway, or by inhibiting the sequestration of MBNL in CUG-foci by steric blockade. PMO ASOs composed of 25 CAG repeats were among the first drugs to show substantial beneficial effects when administrated locally in the muscles of HSA^LR^ mice [87]. In vitro, PMOs can both invade and disrupt the highly stable CUG-RNAs and MBNL1 complexes as well as prevent MBNL1 sequestration. Three weeks after local injection and electroporation of (CAG)25 PMOs into the skeletal muscles of HSA^LR^ mice, the liberation of MBNL was sufficient to improve aberrant alternative splicing and significantly reduce myotonia. *DMPK* mRNAs were not cleaved following PMO ASO treatments despite accelerated decay and translation being observed, likely because of lower nuclear retention. Muscle injections of short (8 or 10 mer) ASOs with LNA chemistries to inhibit MBNL1 and CUG-expanded (CUGexp) RNAs interactions have also been investigated in HSA^LR^ mice [88]. Compared to 25 CAG PMOs, CAG all-LNA ASOs are approximately three times smaller and have a negative charge, but nonetheless, showed similar properties in reducing myotonic discharges and correcting splicing alterations in vivo. Although steric-blocking ASOs that are locally injected into muscles have shown promising results in DM1 mice, they may have insufficient potency for being administered systemically without a carrier. Indeed, it has been reported that subcutaneous injections of CAG LNA/2′-O-Me chimeric ASOs with a PS backbone in HSA^LR^ mice, despite accumulating in tissues, do not even partially reverse splicing alterations at high doses, unlike intramuscular injections [36]. Intravenous injections of PMO ASOs that promote exon skipping have been used in clinical trials to successfully treat DMD [89]. Antisense therapy for DMD is favored by enhanced cell permeability caused by the lack of dystrophin in muscle fibers. In contrast, cell membrane integrity is not compromised in myotonic dystrophy, which may further hinder systemic ASO delivery for this neuromuscular disorder [90]. Furthermore, intranuclear ASOs that act by steric hindrance must be used in high concentrations to cover the major portion of CUG repeats [54]. The dose administrated in patients would probably need to be titrated according to the length of the CTG expansions. 

### 5.2. ASOs with Catalytic Activity for Degrading CUG-Expanded RNAs

ASOs with catalytic activity may be required in smaller quantities and, as such, represent an alternative strategy for efficient targeting while limiting secondary effects. Following an extensive screening in cell lines derived from DM1 patients and DM300 mice, a (CAG)7 ASO with 2′-O-Me and PS backbone modifications has been shown to decrease *DMPK* transcripts in vitro by up to 90% [92]. The authors reported that ASOs with CAG repeats preferentially decrease the expression of transcripts with CUG expansions rather than wild-type *DMPK* mRNAs, likely because of the increased availability of binding sites in the former. As ASOs fully modified with 2′-O-Me do not activate RNase H, the underlying mechanisms causing degradation in these experiments are not clear. Nonetheless, (CAG)7 2′-O-Me PS ASOs injected into the lower limb muscles of HSA^LR^ mice have been shown to reduce CUG-expanded transcript expression by up to 50%, which was sufficient to alleviate mis-splicing events in this tissue. It was further reported that 2′-O-Me ASOs with fewer than five CAG repeats do not reduce the quantity of CUG-expanded *DMPK* mRNAs in DM500 mouse myoblasts, suggesting that a minimal repeat length may be required with some chemistries [55]. For increasing ASO potency, chimeric CAG sequences made with 2′-MOE or LNA ends to improve nuclease resistance as well as a central gap sensitive to RNase H have been designed [91]. An ASO with 2′-MOE terminal sequences of three nucleotides and a central region of eight PS nucleotides has been shown to degrade up to 80% of CUG-expanded transcripts in vitro compared with 25% for mixmer analogs with gaps of no more than three nucleotides that prevented the binding of RNase H. Therefore, RNase H activity is an important mechanism contributing to nuclear *DMPK* degradation by gapmer ASOs. Following intramuscular injection in the EpA960/HSA-Cre mouse model, 2′-MOE-(CAG)14 gapmers degraded approximately 50% of CUG-expanded transcripts and partially corrected mis-splicing events, in contrast with mice injected with a 2′-MOE-(CTG)14 control ASO [91]. PS-modified ASO gapmers with 2′-O-Me terminal sequences efficiently reduced CUG-expanded *DMPK* mRNAs in DM500 mouse myotubes, unlike similar pure DNA CAG-ASOs, highlighting the need to stabilize the ASOs to allow a significant activation of the RNase H pathway [55]. Another in vitro study has shown that CAG LNA and 2′,4′-BNA^NC^[NMe] gapmers are more effective at lower doses than analogs targeting *DMPK* mRNAs outside the repeats, which may be explained by higher availability of binding sites for CAG ASOs [37]. However, CAG-ASOs were less potent at higher doses, with a residual quantity of more than 15% of *DMPK* mRNAs remaining, possibly because of steric hindrance caused by the high-affinity binding of MBNL for the repeats or because of RNA secondary structures.

## 6. Systemic Therapy for DM1 with Naked ASOs

### 6.1. Systemic ASOs Administration for Targeting Skeletal Muscles

The previously mentioned studies required direct intramuscular ASO injection followed by electroporation, which would be unsuitable for systemic treatments of human patients. In addition, tissue damage following intramuscular injections may result in the expression of embryonic isoforms caused by tissue regeneration and may mitigate the beneficial effects of the treatment being investigated [91]. An important step was taken by Wheeler et al., who showed that subcutaneous injections of chimeric ASOs composed of 2′-MOE ends with a central gap of unmodified nucleotides targeting various regions of *ACTA1* transcripts reduced CUG-expanded RNAs up to 80% in HSA^LR^ mice and almost eliminated myotonia [95]. One year after the last dose, the reduction was approximately 50% for the ASO candidate 445236, indicating that it might be feasible to use this approach in a clinical setting. Although other studies with 2′-MOE gapmers achieved significant knock-down in the liver, muscle tissues have been challenging to target with this chemistry. Since RNase H degradation occurs primarily in the nucleus, it has been suggested that CUG-mRNAs are highly sensitive to these mechanisms compared to other targets because of their long nuclear residence time [95]. The cEt gapmer IONIS 486178 that targets human, mice, and monkey *DMPK* mRNAs was shown to possess high potency and good tolerability following systemic administration in multiple in vivo models [63]. Indeed, IONIS 486178 is homologous to a highly conserved sequence in the 3′ UTR of the *DMPK* mRNA and was reported to reduce endogenous *Dmpk* transcripts in the muscles by up to 90% in wild-type mice and 70% in cynomolgus monkeys. Importantly, we have shown that subcutaneous injections of IONIS 486178 at 25 mg/kg in DMSXL mice destroy more than 70% of CUG-RNA foci, improve muscle strength, and correct muscle fiber immaturity [66]. Systemic administration of the same doses of IONIS 486178 in DM200 mice also significantly improved myotonia [67]. Taken together, these data support the notion that inhibiting *DMPK* mRNA expression is a promising strategy for alleviating muscular symptoms in DM1 and is compatible with systemic delivery. It has recently been shown that treating HSA^LR^ mice with a 2′-MOE gapmer ASO targeting CUG-expanded *ACTA1* transcripts efficiently corrects mis-splicing alterations for up to three months but does not reduce muscle fatigue following exercise [96]. On the other hand, the same antisense treatment together with an exercise regimen of treadmill walking had a significant effect on muscle fatigue. Hence, a combination of other therapies with an ASO treatment may have potential benefits for broader symptoms of the disease.

### 6.2. Systemic ASO Administration for Targeting Cardiac Pathology

The heart is another important, yet challenging, target of antisense therapy for myotonic dystrophy. Indeed, heart diseases are a major cause of sudden death in patients with DM1. First-degree atrioventricular block is the most common conduction disorder and other defects, including bundle branch blocks and prolongation of the QRS interval on surface electrocardiogram (ECG), are frequent with increased disease duration or severity [108]. Cardiac histopathology features fibrosis, fatty infiltration, and cardiomyocyte hypertrophy. Systemic administration of the ASO IONIS 486178 in DM200 mice has been reported to correct cardiac conduction defects, such as ranging atrial arrhythmias and heart blocks of varying degrees, in addition to reducing fibrosis [67]. A 50% reduction of the toxic transcripts in the heart restored connexin 40 protein expression and corrected *Atp2a1* exon 22 exclusion, which was related to the correction of cardiac electric activity. It is noteworthy that the reported *DMPK* mRNA silencing by unconjugated ASOs is generally lower in the heart (30–60%) than in muscles (50–90%) following systemic administration [63,64,66,67]. Increased ASO retention in the endosomes of cardiomyocytes compared to skeletal muscle cells may contribute to this phenomenon [109]. Given this, the development of strategies to enhance ASO potency in the heart is crucial for treating cardiac pathologies caused by myotonic dystrophy. 

## 7. Cell-Penetrating Peptide ASOs for DM1

Despite the promising results of unconjugated ASOs in preclinical mouse models, low concentrations of the chemically modified ASO IONIS-DMPKRx in muscle tissues following injections in humans indicate that, beyond chemical modifications, further strategies are required to improve potency in targeted organs (Figure 2). One main challenge for antisense therapy is to enhance cellular uptake. Negatively charged nucleic acids and neutrally charged PNAs and PMOs exhibit a low capacity to penetrate the bi-lipidic plasma membrane, which is impermeable to large polar molecules. Cell-penetrating peptides (CPPs) are being extensively investigated to overcome this challenge as they are often composed of positively charged amino acids, such as lysine or arginine, which facilitate translocation across the membrane [110]. Neutrally charged PMO or PNA ASOs are ideal cargos for CPPs to avoid electrostatic interactions. Although the injection of naked CAG-PMOs requires local administration and electroporation, their conjugation with an arginine-rich CPP (K-peptide) enables them to have a significant effect on MBNL1 release from foci and mis-splicing corrections in addition to correcting myotonia to near-normal levels following intravenous injections at 30 mg/kg in HSA^LR^ mice [98]. Another arginine-rich peptide, Pip6a, has been identified as an ideal candidate for the systemic delivery of PMOs to restore the cardiac expression of dystrophin in a mouse model of DMD [111]. Pip6a is composed of a hydrophobic core flanked on each side by polycationic arginine-rich domains and may be covalently linked to PMO by an amide linkage at the 3′ end of the ASO. Conjugation of CAG-PMO ASOs with Pip6a corrected myotonia and transcriptomic alterations in HSA^LR^ mice following intravenous injection at a dose of 12.5 mg/kg, while the unconjugated ASO did not affect mis-splicing events common in this mouse model [99]. In addition, conjugation of (CAG)5 PS 2′-O-Me-modified ASOs to the amphipathic peptide PepFect14 (PF14) was shown in vitro to increase ASO translocation to the nucleus in muscle cells and enables greater correction of mis-splicing compared to ASOs conjugated to a nona-arginine CCP or administered without a carrier, indicating that CCP conjugation can also impact intracellular distribution [54]. A main drawback for the clinical use of CPPs is the lack of cell type specificity and their potential toxicity at high doses, possibly caused by their cationic properties [112,113]. CCP-conjugated ASOs have indeed been reported to cause kidney tubular degeneration in non-human primates [114].

## 8. Lipid-Conjugated ASOs Targeting Skeletal and Cardiac Muscles

Improving tissue-specific delivery of ASOs is important to achieve a significant therapeutic effect in humans while, at the same time, delivering drug doses low enough to limit potential toxic effects. Ligand-conjugated antisense (LICA) is a promising strategy in which the ligands bind to receptors usually expressed by specific cell types and subsequently promote their uptake by endocytosis. Notably, triantennary N-acetyl galactosamine (GalNAc3) covalently linked to 2′-MOE gapmers has been used to reduce apolipoprotein(a) expression and has been investigated in a phase 2 clinical trial for treating patients with hyperlipoproteinemia and other cardiovascular diseases (NCT03070782). GalNAc3-conjugated ASOs are recognized by asialoglycoprotein receptors expressed specifically by hepatocytes and have around 20-fold higher potency in the liver than unconjugated ASOs [115,116], showing the potential of LICA to enhance specific cell uptake. Fatty acid transport by plasma albumin or as triglycerides by lipoproteins and oxydation are essential for sustaining muscle contractile activity. The conjugation of different ASOs targeting multiple mRNAs such as *DMPK*, *Cav3*, *CD36*, and *Malat-1* with fatty acid chains increases their potency 3- to 7-fold in mouse muscles, with a greater improvement when the fatty acid chain length is between 16 to 18 carbons [101]. The lipid conjugate is usually linked to a phosphodiester group of the ASO by a hexylamino (HA) linker, which is cleaved following endocytosis and releases the intact ASO [102]. Palmitic acid (C16)-conjugated ASOs exhibit 200-fold stronger binding to albumin than unconjugated ASOs and display an increased affinity for low-density lipoprotein (LDL) and high-density lipoprotein (HDL), resulting in higher retention in the blood and reduced secretion in urine [117]. The fatty acid conjugate also promotes ASO transport across the endothelial barrier as well as their accumulation in the interstitium over 24 h, increasing the short-term exposure of the drugs to muscles despite a relatively small intracellular accumulation. Intriguingly, the reported intracellular accumulation of palmitic acid (C16)-conjugated ASOs is modest compared to the observed increase in potency [117]. It is, thus, likely that the lipid conjugate also enhances ASO activity by acting on uptake pathways and influencing their intracellular distribution [118]. Indeed, it has been observed that lipid conjugates promote ASO release from endosomes [119]. The hydrophobic lipid conjugates tocopherol and cholesterol have also been shown to increase ASO potency in mouse myocytes, although tocopherol is more effective when administrated intravenously than subcutaneously whereas cholesterol is toxic at high doses [102].

IONIS has designed palmitoyl (C16)-conjugated ASOs that target CUG-expanded mRNAs to optimize muscle and cardiac tissue targeting in DM1 therapy. It has been reported that the conjugation of palmitic acid with IONIS 486178 improves *Dmpk* degradation up to 4-fold in the mouse quadriceps and heart [101,102]. Accordingly, systemic injection of a C16 fatty acid conjugated to an ASO targeting *ACTA1*-CUG transcripts (IONIS-992948) in the bichromatic reporter HSA-TR mouse model followed by serial in vivo spectroscopy showed a 2-fold improved correction of mis-splicing in muscles compared to unconjugated ASOs with the same sequence [78]. We noted that subcutaneous injection of 25 mg/kg of IONIS 486178 conjugated to a C16 palmitic acid via an HA linker (IONIS-877864) in DMSXL mice results in a 90% reduction in human *DMPK* mRNAs in muscles compared to about 60% for the same dose of the unconjugated ASO [64]. Yadava et al. investigated the effect of IONIS-877864 on muscle regeneration by inducing muscle damage with BaCl_2_ in DM200 mice [103]. DM200 mice induced to express the CUG-expanded transgene featured impaired muscle regeneration and decreased muscle fiber maturation following muscle injury. Systemic treatment with IONIS-877864, but not a control LICA, improved these features by increasing the number of muscle satellite cells and regenerating muscle fibers in induced DM200 mice. We also observed that IONIS-877864 decreases *DMPK* mRNA expression in the heart by 78% compared to 65% for the unconjugated ASO following systemic administration in DMSXL mice [64], confirming the relevance of fatty-acid conjugated ASOs for treating cardiac tissues as well. 

## 9. Antibody-Conjugated ASOs to Enhance Cell Uptake via the Transferrin Receptor

ASO conjugation to monoclonal antibodies is another strategy to enhance receptor-mediated cell uptake. Transferrin receptor 1 (TfR1), which mediates the endocytosis of iron-bound transferrin molecules in cells, has been known for decades as an interesting target for mediating the internalization of oligonucleotides [120]. Following internalization by clathrin-dependent endocytosis and liberation of its cargo by the acidic pH of endosomes, TfR1 is quickly recycled to the surface and becomes available for further transport. Combining ASOs with transferrin ligands is not ideal because of the large size of transferrin (79.5 kDa) and potential binding competition with endogenous transferrin that would decrease iron uptake. Instead, oligonucleotide conjugation with monoclonal antibodies or a fragment antigen-binding region (Fab) targeting Tfr1 outside the binding site is favored. TfR1 is broadly expressed in the body, including in muscle and cardiac tissues for the synthesis of myoglobin, and has been shown in vivo to improve conjugated siRNA delivery to these tissues [121]. 

Thereby, Dyne Therapeutics has developed the FORCE^TM^ platform, which is composed of an ASO combined with a Fab targeting TfR1 via a linker that is stable in serum but that allows cargo release from endosomes. Conjugation of an exon-skipping PMO with a mouse-specific TfR1-targeting Fab of the FORCE^TM^ platform was recently reported to significantly enhance delivery to the muscles and the heart following intravenous administration in a mouse model of DMD [122]. The potency of the FORCE^TM^ DM1 ASO (DYNE-101) targeting *DMPK* mRNAs was investigated in DMSXL mice expressing human TfR1 (hTfR1/DMSXL mice) [123,124]. The administration of a single intravenous dose of 10 mg/kg of the FORCE^TM^ DM1 ASO to hTfR1/DMSXL mice reduced the level of human *DMPK* mRNAs with expanded CUG repeats by around 40–50% in skeletal muscles as well as in the heart for four weeks. The same FORCE^TM^ technology rescued mis-splicing and myotonia to near-normal levels in HSA^LR^ mice within 14 days after administering a single dose, further supporting the therapeutic potential of the compound [124]. In non-human primates, monthly doses of 10 mg/kg of DYNE-101 achieved between 30 and 70% of wild-type *DMPK* mRNA destruction in skeletal, smooth, and cardiac muscle tissues, while displaying no toxicity at higher doses [124]. Furthermore, drug delivery to the nucleus was validated by subcellular fractionation followed by gene analysis [123]. Following these encouraging preclinical results, Dyne Therapeutics has recently announced the launch of a clinical trial (NCT05481879) to evaluate the safety and pharmacologic properties of Dyne-101 in patients with DM1. 

Furthermore, Avidity Biosciences has developed a TfR1 monoclonal antibody conjugated to a siRNA targeting human *DMPK* mRNAs (AOC 1001). Avidity has reported that in vitro treatment of DM1 myotubes partially corrected alternative splicing errors, while a single dose of 2 mg/kg administrated in non-human primates reduced *DMPK* mRNAs by ~75% over 12 weeks in skeletal muscles [125]. In addition, it was shown that AOC 1001 accumulates in the tissue at nanomolar concentrations in a dose-dependent manner. These results, showing efficient potency of AOC 1001 without causing toxicity, led to the launch of a phase 1/2 clinical trial in 2021 (NCT05027269) to evaluate the intravenous administration of the drug in adult patients with DM1 [126].

## 10. Improving ASO Delivery in the Brain

Neurological symptoms caused by DM1 contribute significantly to poor quality of life in patients and include excessive daytime somnolence, executive dysfunction, as well as verbal and visual memory impairments. The brain is probably the most challenging tissue for drug delivery because of the blood-brain barrier (BBB), which restricts the movement of numerous large molecules, including most ASOs. Notably, we have reported that both the cEt gapmer (ISIS 486178) and C16-HA ASOs (IONIS-877864) do not reach the brain following subcutaneous injection [64,66]. To overcome this limitation, we recently injected IONIS 486178 by intracerebroventricular injection in DMSXL mice and succeeded in destroying up to 70% of *DMPK* mRNAs in multiple brain areas [65]. Intrathecal administration of the ASO nusinersen has been approved by the FDA for clinical use to treat SMA, further supporting the feasibility of this approach in humans [127]. We have observed that *DMPK* mRNA downregulation in the brain of DMSXL mice following an intraventricular bolus injection of IONIS 486178 was maintained at 30–50% for twelve weeks [65]. Intrathecal administration of ASOs for the brain could, thus, potentially be administered at the same time as a subcutaneous or intravenous injection for treating cardiac and muscle tissues. Alternatively, the recent development of new compounds that can cross the BBB may allow brain targeting without such an invasive treatment. TfR1 is expressed by the microvascular endothelial cells of the BBB to transport iron-loaded transferrin from the blood to the brain [128]. Thus, the conjugation of ASOs with antibodies targeting TfR1 may promote ASO transcytosis across the BBB by receptor-mediated endocytosis. It has been reported that biotinylated radiolabeled PNAs linked to a streptavidin-conjugated monoclonal antibody targeting the mouse transferrin receptor are distributed in the brain following intravenous injection, unlike the unconjugated ASO [129]. Other strategies, such as conjugation to CPPs, loading into extracellular vesicles, and encapsulation in glucose-coated polymeric nanocarriers, have been reported to deliver ASOs to the mouse brain following systemic administration [130,131,132]. Significant silencing of target RNAs in the brain has also been achieved following intravenous coadministration of an oligonucleotide and angubindin-1 to increase the permeability of the BBB [133]. Older DM1 patients often manifest structural alterations of the brain related to disease duration such as ventricular dilation, cortical atrophy, and loss of grey volume [134,135,136]. Aberrant alternative splicing of *MAPT* is thought to lead to the accumulation of aggregated phosphorylated Tau and the formation of neurofibrillary tangles in the brains of DM1 patients [137]. CNS symptoms resulting from neurodegeneration in DM1 would, thus, likely benefit from preventive antisense therapy. We have observed that intracerebroventricular injection of IONIS-486178 in neonatal DMSXL mice prevents the development of abnormal behaviors, suggesting that this treatment may be promising for treating neurodevelopmental disorders in early-onset forms of the disease [65].

Both the large size and the charge of ASOs are important drawbacks for efficient delivery across the cell membrane and the BBB [138]. In vitro experiments have shown that the combination of short uncharged MPγPNA ASOs with only two CAG repeats with terminal pyrene moieties increases binding affinity through the formation of π–π interactions and enhanced binding cooperativity [139]. Further experiments showed that these (CAG)2 MPγPNAs conjugated to aromatic moieties specifically discriminate CUG-expanded mRNAs from normal length transcripts and prevent their interaction with MBNL1, making it an interesting candidate for treating DM1. NeuBase Therapeutics has recently developed, through its Peptide-Nucleic Acid Antisense Oligonucleobase Platform (PATrOL™), a novel PNA drug (NT-0231.F) that targets CUG repeats with a proprietary chemistry, allowing broad biodistribution in tissues. NeuBase Therapeutics recently divulged preclinical data showing rapid uptake and prolonged retention of NT-0231.F in the muscles, the heart, and even the brain following systemic administration in BALB/c mice [140]. They reported that a single intravenous injection or multiple subcutaneous injections of NT-0231.F in HSA^LR^ mice reduce the number of nuclear foci, rescue splicing defects, and improve muscle relaxation after four weeks, further supporting its potential for clinical translation [141]. Notably, NeuBase reported that their PNA drug crosses the BBB following systemic injection in non-human primates and accumulates in the CNS, including in deep brain structures [142]. One week after a single injection, the drug remains in the brain, with a 2-fold enrichment in some brain regions occurring through drug redistribution. Despite important challenges to achieving efficacious ASO delivery to the CNS, these recent advances hold great potential for treating brain defects in myotonic dystrophy. 

## 11. Safety and Tolerability of *DMPK*-Targeting Antisense Therapy 

### 11.1. Sequence-Dependent Toxicity of DMPK Targeting ASOs

ASO toxicity can depend on the nucleotide sequence and the results of their hybridization with target and off-target RNAs or can be independent of the sequence and rather be determined by the chemical properties of the ASO itself. ASOs composed of CAG repeats often allow specific targeting of mutant *DMPK* mRNAs in DM1 cells, enabling the production of functional DMPK proteins from the wild-type allele. A main concern is the potential existence of multiple other transcripts containing CUG repeats which can be potential off-targets of ASOs with degradation activity. The effect of different CUG-targeting ASOs was investigated on candidate transcripts containing at least six CUG triplets in mice [91,92]. There was no major expression alteration or only a slight decrease in some cases following high ASO doses. These results suggest that RNAs with a large repeat expansion usually have a better affinity for the ASOs than other non-target transcripts. In contrast, another in vitro study found that BNA^NC^ gapmers targeting *DMPK* outside of the repeats preferentially degraded nuclear-retained CUG-expanded *DMPK* mRNAs, while CAG BNA^NC^ gapmers degraded both the transcripts from the normal and pathogenic expanded alleles in addition to other RNAs with CUG repeats [37]. 2′-MOE-(CTG)14 and –(CAG)14 gapmers electroporated into mouse muscles have been reported to induce histopathological changes associated with regeneration, unlike the mock control [91]. It is possible that repeat oligonucleotides exceeding a certain length adopt three-dimensional structures with immunostimulatory or aptameric properties [143,144]. CUG-expanded RNAs may hybridize to the template strand during transcription to form an R-loop, and thereby increasing CTG trinucleotide instability [145]. This raises concern that CAG ASOs could form a D-loop and increase the expansion of CTG repeats, leading to the progression of symptom severity. However, the injection of CAG-LNA ASOs in transgenic mice carrying the human *DMPK* gene with about 800 CTG repeats halted genetic instability, most likely by reducing the formation of R-loop structures by the CUG-expanded RNAs [146].

Targeting the regions flanking the CTG repeats is another common strategy for treating DM1 but is not specific to CUG-expanded mRNAs and involves the degradation of wild-type *DMPK* mRNAs. This may aggravate the haploinsufficiency caused by mutated *DMPK* transcripts that are retained in the nucleus and not translated. DMPK is a serine/threonine protein kinase homologous to the Rho-associated kinases and whose function is not well known. DMPK regulates the myosin phosphatase target subunit (MYPT1) and is involved in actomyosin contraction and cytoskeletal rearrangement [147]. Phospholemman and phospholamban, which are respectively involved in ion transport and calcium uptake in cardiomyocytes, have also been identified as substrates of DMPK in vitro, although the physiological relevance remains to be determined [148,149]. DMPK is mainly expressed in skeletal and smooth muscles, in the intercalated discs of cardiac muscles as well as in the choroid plexus and the synaptic regions of the developing brain [150,151,152]. Homozygous *Dmpk* knock-out mice manifest late-onset mild myopathy consisting of small variable changes in the size of the head and neck muscle fibers as well as microstructural changes in myofibrillar organization [153,154]. DMPK could, thus, be implicated in muscle structure and integrity. In addition, DMPK regulates the intracellular trafficking of insulin and insulin-like growth factor-1 (IGF-1) receptors, which is coherent with decreased insulin signaling in the muscles of homozygous *Dmpk* knock-out mice [155]. Nonetheless, insulin resistance in DM1 patients is thought to be caused mainly by aberrant production of the fetal isoform of the insulin receptor in adult muscle tissues and by altered post-receptor signaling [156,157]. Conduction defects, like those of DM1 patients, such as age-dependent first-degree atrioventricular blocks, have been reported in heterozygous adult *Dmpk* knock-out mice [158,159]. Another study showed that there are fewer β1-adrenergic receptors at the cardiac sarcolemma of *Dmpk* knock-out mice, suggesting that DMPK protein plays a role in the regulation of membrane trafficking [160]. *Dmpk* knockout mice have also been reported to feature increased CELF phosphorylation and nuclear localization [161]. To address the potential effects of ASOs in aggravating some DM1 symptoms by further reducing DMPK protein expression, cardiac and muscle functions were evaluated in heterozygous *Dmpk* mice in which the expression of *Dmpk* mRNAs in these tissues was reduced by 90% by the injection of the ASO IONIS 486178 [97]. In contrast with previous studies, DMPK genetic and antisense knockdown in two mouse models with different genetic backgrounds and at various ages resulted in neither altered muscle strength nor myopathy or cardiac conduction defects, although a minor reduction in the cardiac ejection fraction following cardiac pressure overload was observed. It has been suggested that the contrasting results obtained in a previous study may have been caused by the altered expression of the genes flanking *DMPK* by the selection cassette used to generate the *Dmpk* knock-out mouse model [97]. The injection of the same ASOs has also been reported to cause normal apparent muscle histology in multiple mice strains and non-human primates [63]. Furthermore, a 50% reduction in *DMPK* mRNAs in the hearts of cynomolgus monkeys caused by IONIS-486178 did not result in any cardiac conduction defects based on ECG analysis [63]. In a recent study, triple heterozygous knockout mice for *Six5*, *Dmpk*, and *Mbnl* manifested increased myocardial fiber abnormalities and a lower ejection fraction at the adult age, and additional knock-down of *Dmwd* caused ventricular and atrial wall attenuation during development [162]. It is thus possible that the downregulation of DMPK protein can cause pathological effects when combined with the dysregulation of other genes affected by the disease. However, preclinical studies indicate that, overall, the benefits of *DMPK* mRNA destruction by ASOs in DM1 muscle and cardiac tissues greatly surpass the possible risks associated with the reduced expression of the protein. 

### 11.2. Nephrotoxicity and Hepatotoxicity of ASOs

As most ASOs accumulate in high concentrations in the liver and kidneys, varying levels of toxicity in these organs have been reported with some sequence and chemistry combinations, especially at higher doses than those used in a clinical setting [58,163]. Non-specific interactions of polyanionic ASOs with proteins are thought to contribute to hepatic degeneration [164]. Furthermore, kidney dysfunction, such as impaired tubular absorption or increased tubular proteinuria, may be caused by ASOs accumulation into the lysosomes of proximal tubular cells [164]. Blood analyses for biomarkers of kidney and liver damage are thus required during in vivo investigations of ASOs to ensure clinical translation. Proinflammatory and immunostimulant effects of ASO administration have been reported in mice and non-human primates, and signs of inflammation or necrosis in the liver and the kidney should therefore also be assessed by histopathological evaluation [165,166]. Notably, PS-modified ASOs are known to cause blood vessel inflammation and immune cell infiltrates, most likely following interaction with the complement system [164]. ASOs’ potential toxicity in the liver and the kidney is a main challenge for clinical translation, which may be alleviated by enhancing delivery to targeted tissue and reducing the required dosage. 

## 12. Targeting Mechanisms Downstream of RNA Toxicity with Antisense Therapy

### 12.1. AntagomiR and BlockmiR ASOs 

The versatility of antisense therapy makes it possible to target other important downstream mechanisms associated with DM1 symptoms. For example, the striking recapitulation of mis-splicing of the chloride channel 1 (*Clcn1*) gene specific to skeletal muscles and myotonia in homozygous *Mbnl1* exon 3 knock-out mice (*Mbnl1ΔE3/ΔE3* mice) confirms that the loss of function of this RBP is a major cause of the muscular symptoms manifested by DM1 patients [167]. Increasing the expression of MBNL1 in HSA^LR^ mice alleviates DM1 symptoms, which supports the therapeutic potential of this approach [168,169]. Two micro-RNAs (miRNAs), miR-23b and miR-218, have been identified to downregulate MBNL1/2 mRNAs in human cells and are highly expressed in mouse skeletal muscles, hearts, and brains [104]. Therefore, ASOs with a PS backbone at the terminal ends, 2′-O-Me modifications, and conjugated to a cholesterol moiety to enhance cell uptake in muscle tissues were designed to act antagonistically to miR-23b and -218. Subcutaneous or intravenous administration of these ASOs, called antagomiR-23b and -218, at a total dose of 12.5 mg/kg in HSA^LR^ mice caused an increase in MBNL expression, which was sufficient to improve MBNL-dependent mis-splicing events, muscle histopathology, muscle strength, and myotonia [104,105,106,107]. However, this strategy is limited by the lack of therapeutical effect on CELF-dependent transcriptomic alterations and the possibility that other miR-23b or -218 off-target RNAs are affected. PMO ASOs (blockmiRs) conjugated to a cationic CPP (Pip9b2) were thus designed to specifically inhibit the binding of miR-23b to the 3′ UTR of *MBNL* mRNAs by steric hindrance [100]. BlockmiRs were injected via the tail vein in HSA^LR^ mice and showed a therapeutic potential similar to that of antagomiRs at a similar dose in terms of an improvement in grip strength. The promising results of this strategy led Arthrex Biotech to develop the drug ATX-01 using the proprietary ENTRY^TM^ platform to enhance delivery in tissues to increase MBNL expression by acting on miRNAs [170].

### 12.2. Splice-Switching ASOs for Correcting Myotonia 

Antisense drugs acting by steric hindrance can also be used to correct specific splicing defects by exon skipping. MBNL promotes the splicing of the *CLCN1* exon 7A, but the aberrant inclusion of this exon in adult DM1 tissues causes a codon frameshift and premature translation termination of mRNAs [171,172]. In contrast with full-length ClC-1 protein, truncated isoforms including the exon 7A do not localize at the muscle sarcolemma and do not exhibit channel activity. Therefore, PMO ASOs have been designed to target the *Clcn1* exon 7 pre-mRNA 3′ splice site and were injected and electroporated locally in the muscles of HSA^LR^ and *Mbnl1ΔE3/ΔE3* mice [93]. In both models, ASOs treatments restored ClC-1 expression at the sarcolemma and corrected myotonia. To enhance PMO local delivery, ASOs were injected together with gas-entrapping liposomes, called bubble liposomes, in the tibialis anterior muscle of HSA^LR^ mice and exposed to ultrasound [94]. This process is thought to facilitate oligonucleotides’ penetration in cells by temporarily decreasing membrane permeability [173]. Indeed, PMO injected in combination with bubble liposomes and ultrasound application achieved a greater correction of *Clcn1* splicing and a reduction of myotonia [94]. These results support that aberrant splicing of *CLCN1*, instead of downregulation of the protein, is responsible for myotonic discharges, and constitute a valuable target for motor impairment in DM1.

## 13. Alternative RNA-Based Strategies for Treating DM1

### 13.1. Short Interfering RNAs 

Aside from ASOs, other RNA-based therapeutics have been investigated for targeting CUG-expanded mRNAs. For instance, siRNAs can be delivered to cells as RNA duplexes where they interact with the multi-protein RNA-induced silencing complex (RISC). A passenger strand is designed to have less favorable interactions with RISC and is discarded, while the other active guide strand binds to the target CUG-expanded mRNA, leading to its cleavage by the protein AGO2 in the case of perfect pairing. The injection and electroporation of CAG·CUG siRNA duplexes in the muscles of HSA^LR^ mice have been shown to degrade up to 80% of transgene expression and exhibit stronger potency than fully modified PMO and 2′-O-Me ASOs, as a ~10-fold lower molar dose was required to achieve a comparable therapeutical effect [174]. These results support that siRNAs enable the efficient degradation of nuclear CUG-expanded transcripts even though RNA interference is thought to act mainly in the cytoplasm [175]. Like ASOs, siRNAs can be chemically modified to enhance their stability for systemic delivery. However, possible modifications are more limited in the case of siRNAs as they can alter interactions with RISCs. Intravenous injection of adeno-associated viruses (AAVs) carrying miRNA-based hairpin vectors that target *ACTA1* transcripts reduced CUG-expanded RNAs by 60–95% in HSA^LR^ mice, in addition to reversing myotonia, muscle histopathology, and the formation of nuclear foci [176]. The ongoing clinical investigation of the TfR1 antibody-conjugated siRNA drug AOC 1001 developed by Avidity Biosciences confirms the translational potential of this class of RNA therapeutic for treating DM1.

### 13.2. Single-Guide RNA and Cas9 Ribonucleoproteins

Another strategy involves the systemic administration of AAVs in adult HSA^LR^ mice to induce the prolonged co-expression of a nuclease-dead Cas9 protein and a single-guide RNA targeting the CUG repeats [177]. This strategy achieved almost a 60% reduction in CUG-expanded mRNAs in quadricep muscles and the correction of myotonia to near-normal levels. Because the expression of non-self Cas9 protein for an extended period is immunogenic, it is necessary to co-administer an immunosuppressant to achieve a therapeutical effect. 

### 13.3. Hammerhead Ribozymes

Hammerhead ribozyme is another class of RNA that has been investigated for downregulating RNA toxicity caused by CUG-expanded transcripts. Ribozymes feature an RNA motif with a catalytic activity that can cleave the phosphodiester bonds of specific targets. In vitro, nuclear-retained hammerhead ribozyme achieved almost a 60% reduction in CUG-expanded *DMPK* mRNAs in DM1 myoblasts [178]. However, the clinical translation of ribozymes has been hampered by the challenges in optimizing in vivo delivery, making other RNA therapeutics more attractive [179]. 

### 13.4. Small Nuclear RNAs

U7 small nuclear RNA (U7 snRNA) is involved in the processing of the 3′ end of histone pre-mRNAs, unlike other snRNAs that regulate alternative splicing [180]. In the search for a new therapeutic tool for DM1, U7 snRNA has been modified by replacing the original sequence complementary to histone pre-mRNA with antisense CAG repeats [181]. Additionally, the original Sm binding motif was substituted by a canonical sequence derived from the predominant spliceosomal U small nuclear RNAs to improve nuclear localization. Expression of the CAG-U7 snRNAs reduced *DMPK* mRNAs by more than 70% in DM1 cells, restored myogenic differentiation, and partially corrected alternative splicing defects [72,181]. U7 snRNAs have the advantage of being protected from degradation by forming small nuclear ribonucleoprotein (snRNP) complexes and accumulating in the nucleus while being non-toxic and non-immunologic for cells. Encapsidation of vectors containing U7 snRNA sequences promoting exon skipping in AAVs and their systemic delivery in mouse and dog models of DMD have been reported to alleviate the dystrophic phenotype in multiple studies [182,183,184,185,186]. Importantly, the stable expression of the U7 snRNA allowed partial restoration of full-length dystrophin expression for more than 1 year after the injection [183,184]. A therapy based on AAV U7 snRNA is currently under investigation in a clinical trial in phase 1/2 for treating boys with DMD exon 2 duplication (NCT04240314), supporting the translational potential of this method. Nonetheless, antisense therapy remains the most widely used approach for treating DM1, notably because they can be easily chemically modified to improve in vivo potency and because they are well suited for systemic administration without the use of AAV vectors. 

## 14. Conclusions

The inhibition of CUG-expanded *DMPK* mRNAs and their pathogenic interaction with RBPs by antisense oligonucleotides is the focus of extensive research studies that have shown their strong potential to reverse DM1-like symptoms in multiple transgenic mouse models. However, it is crucial to determine the right combination of nucleotide sequences, chemical modifications, and conjugates to maximize delivery and potency in targeted tissues and achieve significant results in clinical trials. The first clinical trials for DM1 with a conjugated siRNA (NCT05027269) or ASO (NCT05481879), led respectively by Avidity Biosciences and Dyne Therapeutics, are ongoing and will provide important insights into the pharmacokinetics and pharmacodynamics of oligonucleotide drugs specifically designed to increase uptake into cardiac and muscle tissues. The PATrOL™ platform developed by NeuBase Therapeutics, and the drug ATX-01 developed by Arthrex Biotech also showed promising potential in preclinical DM1 mouse models and are likely to proceed to phase 1 and 2 clinical trials in the years to come. In contrast, FDA-approved drugs repurposed for treating DM1 have already reached phase 2 and 3 trials for Tideglusib (NCT03692312), Metformin (2018–000692-32), and Mexiletine (NCT04624750). These drugs are closer to being accepted for clinical use to alleviate the symptoms of DM1 patients, notably because the safety and pharmacology of these drugs are already well established. However, targeting the central toxic gain-of-function is likely to have broader beneficial effects than regulating downstream mechanisms with repurposed FDA-approved drugs and significantly enhance the quality of life of patients. Achieving optimal ASO delivery to the muscles, heart, and brain will be an important step for the treatment of neuromuscular diseases.

## Figures and Tables

**Figure 1 ijms-23-13359-f001:**
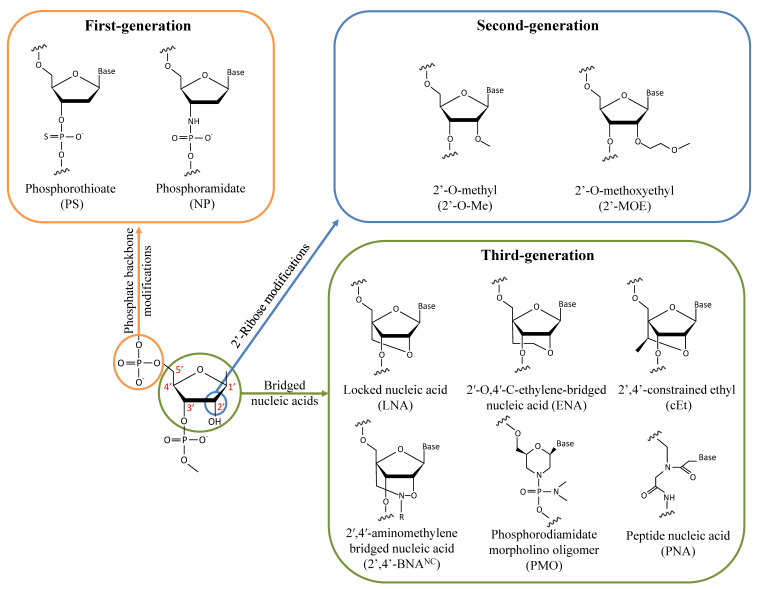
Chemical modifications of the sugar-phosphate backbone of ASOs that have been investigated for treating DM1.

**Figure 2 ijms-23-13359-f002:**
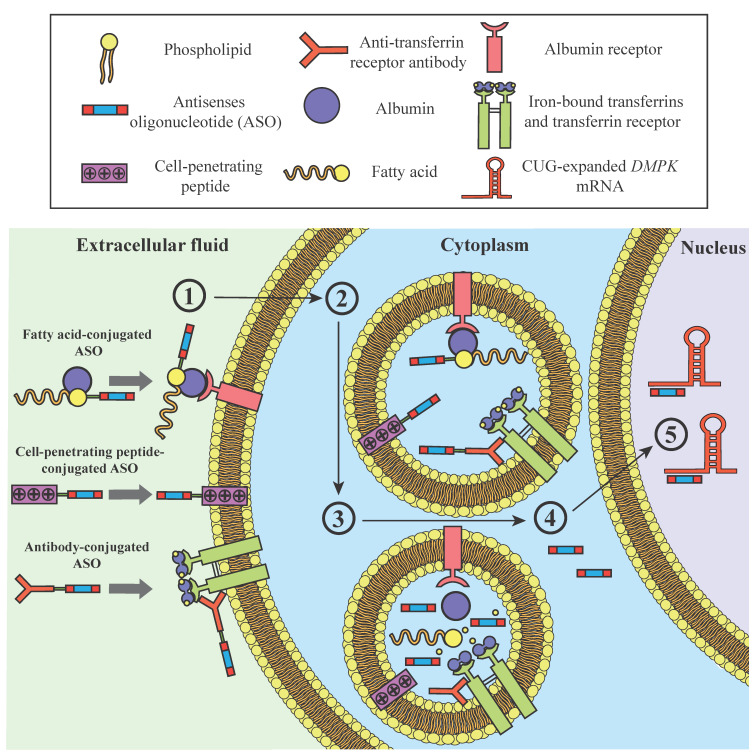
Strategies to enhance antisense oligonucleotide (ASO) intracellular delivery. ASOs may be conjugated to a fatty acid chain, enhancing its affinity to albumin and other serum proteins, to a cell-penetrating peptide (CCP), or to an anti-transferrin receptor 1 (TfR1) monoclonal antibody to promote cell uptake. (1) Conjugated ASOs are localized at the cell membrane, either because of the conjugate binding to specific cell receptors or because of interactions of the CCP with the cell membrane. (2) ASOs are then internalized in early endosomes by endocytosis. (3) An increase in pH in endosomes promotes cleavage of the ASOs’ linker and their release from the conjugates. (4) A small portion of ASOs escape the endocytic vesicles and cross the nuclear pores to the nucleus, (5) where they can target CUG-expanded RNAs.

**Table 1 ijms-23-13359-t001:** Summary of antisense strategies investigated for treating DM1.

Strategy	Molecule	Mechanism	Target	Administration Route	In Vivo Model	Biological Readout	References
Full-length modified ASO	PMO-CAG	Steric hindrance	CUG sequence	Intramuscular injection	HSA^LR^	Reduction of foci number, MBNL sequestration, and mis-splicing in muscles; alleviation of myotonia	[55,87,91]
	LNA-CAG chemistries (all-LNA and LNA/2′-O-Me)	Steric hindrance	CUG sequence	Intramuscular injection	HSA^LR^	Reduction of foci number, MBNL sequestration, and mis-splicing in muscles; alleviation of myotonia	[36,88]
	2′-O-Me-CAG	Enzymatic degradation	CUG sequence	Intramuscular injection	DM500, HSA^LR^	Reduction in CUGexp RNA levels, foci number, MBNL sequestration, and mis-splicing in muscles	[55,71,72,73,74,92]
	PMO	Exon skipping	*Clcn1* pre-mRNA exon 7 3′ splice site	Intramuscular injection	HSA^LR^, *Mbnl1ΔE3/ΔE3*	Reduced inclusion of *Clcn1* exon 7, normalization of the channel activity of ClC-1, alleviation of myotonia	[93,94]
Gapmer	2′,4′-BNA^NC^	RNase H-mediated degradation	*DMPK* 3′ UTR or CUG	-	-	Reduction in CUGexp RNA levels, foci number, and mis-splicing	[37]
	2′-MOE/LNA chemistries (ASO 445236)	RNase H-mediated degradation	*ACTA1* or CUG	Subcutaneous or intramuscular injection	EpA960/HSA-Cre, HSA^LR^	Reduction in CUGexp RNA levels, foci number, and mis-splicing in muscles; alleviation of myotonia	[77,91,95,96]
	cEt chemistries (IONIS 486178)	RNase H-mediated degradation	*DMPK* 3′ UTR	Subcutaneous or intracerebroventricular injection	DMSXL, DM200, *Dmpk*(−/+), wild-type mice and rats, non-human primates	Reduction in *DMPK* mRNA levels in skeletal muscles, hearts, and brains; correction of myotonia, muscle weakness and muscle fiber immaturity; correction of cardiac conduction defects and behavioral abnormalities	[63,65,66,67,97]
CCP conjugation	B/K-peptide-PMO	Steric hindrance	CUG sequence	Intramuscular or intravenous injection	HSA^LR^	Reduction of foci number, MBNL sequestration and mis-splicing in muscles; correction of myotonia	[98]
	Pip6a-PMO	Steric hindrance	CUG sequence	Intravenous injection	HSA^LR^	Reduction of foci number, MBNL sequestration and mis-splicing in muscles; correction of myotonia	[99]
	PepFect14	Steric hindrance	CUG sequence	-	-	Reduction of MBNL sequestration in foci	[54]
	Pip9b2-BlockmiR/ATX-01 (Arthex Biotech)	Steric hindrance	*MBNL1* 3′ UTR	Intravenous injection	HSA^LR^	Increase of MBNL expression, decrease of MBNL-dependent mis-splicing, alleviation of muscle weakness	[100]
Fatty acid conjugation	C16-cEt gapmer (IONIS-877864)	RNase H-mediated degradation	*DMPK* 3′ UTR	Subcutaneous injection	DMSXL, DM200, BALB/c, rats, non-human primate	Improvement of ASO uptake in muscle and cardiac tissues, reduction of foci number and *DMPK* mRNA levels, improvement of muscle strength and regeneration	[64,101,102,103]
	C16-2′-MOE gapmer (IONIS-992948)	RNase H-mediated degradation	*ACTA1*	Subcutaneous injection	TR;HSA^LR^	Improvement of mis-splicing correction in gastrocnemius and lumbar paraspinal muscles	[78]
	Cholesterol-antagomiR-23b/218	MBNL1 overexpression	miR-23b and miR-218	Intravenous or subcutaneous injection	HSA^LR^	Increase in MBNL expression, decrease of MBNL-dependent mis-splicing, alleviation of muscle weakness and myotonia	[104,105,106,107]
Anti-TfR1 antibody conjugation	AOC 1001 (Avidity Biosciences)	Enzymatic degradation	*DMPK* mRNA	Intravenous injection	Non-human primates	Reduction in *DMPK* mRNA levels and mis-splicing	-
	DYNE-101 (Dyne Therapeutics)	Enzymatic degradation	*DMPK* mRNA	Intravenous injection	hTfR1/DMSXL, HSA^LR^, non-human primates	Reduction in CUGexp RNA levels and mis-splicing; correction of myotonia	-
PNA antisense oligonucleobase platform	NT-0231.F (NeuBase Therapeutics)	Steric hindrance	CUG sequence	Intravenous or subcutaneous injection	BALB/c, HSA^LR^, non-human primates	Reduction in foci number, correction of mis-splicing and myotonia	-

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
