# Peer review of "Recent Progress and Challenges in the Development of Antisense Therapies for Myotonic Dystrophy Type 1"

_ijms, 2022, doi:10.3390/ijms232113359_

Round 1
Reviewer 1 Report
The manuscript submitted by Thiéry De Serres-Bérard et al. reviews recent progress in the development of ASO therapies in DM1. The main data are described in this review. It is appreciable to see that the authors have updated their review very recently with data presented during the conference organized by the MDF in early September. The description of the data is clear and well organized. The paper is very well written. Only minor changes are requested.
Comments
Abstract-Line 18- Toxic RNA gain-of-function and gene mis-splicing is also observed in other tissues. Clarify the sentence.
Page 2-Line 76 - Replace “int” by “into”
Page 3- Figure 1 third generation- It is difficult to follow the text with the figure. It is not the same order of citation. Could you please modify Figure 1 to clarify this point. I suggest starting with LNA on the first line and starting with PMO on the second line (first line: LNA-cEt, BNANC, second line: PMO and PNA).
Page 4-Line 132- Could you check if it is BNANC or BNANC in the text.
Page 5- Line 171. Replace [5152] by [51,52]
Page 6- Line 200. Replace [7071] by [70,71]
Page 7- Lines 283 and 286. Replace 84 and 85 by [84] by [85]
Page 8- Line 291. Replace the dot with a comma
Table 1: It is very difficult to follow the text with Table 1. It is not in the same order of citation. There are some references missing from Table 1. The authors need to rearrange the table to make it easier to read.
Page 11- Line 357. Replace 100 by [100]
Page 12- Line 410. Replace [62,63,65101,] by [62,63,65,101]
Page 18- Line 622. One point is missing before Notably
Page 18- Line 638. Replace [9293,137] by [92,93,137]
Page 19- Line 675. Replace [15152] by [151,152]
Page 19- Lines 680 and 681. Specify whether it is the DMPK protein or the DMPK RNA.
Page 20- Line 714. Replace 160] by [160]
Page 20- Line 749. Replace HSALR with HSALR
Page 21- Lines 754 and 756. Replace [174 and [173 by [174] and [173]
Page 21- Line 777. The sentence is not finished. Clarify this point.
Page 21- Line 778. Single is missing a capital letter.
Page 21- Lines 789-791. The link with the previous sentence is not clear. Is it in the right paragraph?
Page 22- Lines 754 and 756. Replace 71,182 by [71,181]
References: 181 and 182 are the same reference. Fix this point.
No acknowledgments?
Reviewer 2 Report
This comprehensive review article discusses ASO's in DM1 and the major positive point is that also addresses the limitations of current DM1 models with respect to ASO therapies.
I have the following two minor comments:
1) Line 60: “myotonia” is a slow/abnormal relaxation of muscle not a prolonged muscle contraction.
2) Line 793-802 on the U7snRNA discussion: adding a few sentences on the preliminary success with U7snRNA in Duchenne muscular dystrophy is useful for this topic. The following paper is a recent example on this approach in DMD:
Wein N, et al. Systemic delivery of an AAV9 exon-skipping vector significantly improves or prevents features of Duchenne muscular dystrophy in the Dup2 mouse. Mol Ther Methods Clin Dev. 2022 Jul 11; 26:279-293. PMID: 35949298.
